# SCALING LAWS FOR THE PRINCIPLED DESIGN, INITIALIZATION, AND PRECONDITIONING OF RELU NETWORKS

## ABSTRACT

In this work, we describe a set of rules for the design and initialization of well-conditioned neural networks, guided by the goal of naturally balancing the diagonal blocks of the Hessian at the start of training. We show how our measure of conditioning of a block relates to another natural measure of conditioning, the ratio of weight gradients to the weights. We prove that for a ReLU-based deep multilayer perceptron, a simple initialization scheme using the geometric mean of the fan-in and fan-out satisfies our scaling rule. For more sophisticated architectures, we show how our scaling principle can be used to guide design choices to produce well-conditioned neural networks, reducing guess-work.

## 1 INTRODUCTION

The design of neural networks is often considered a black-art, driven by trial and error rather than foundational principles. This is exemplified by the success of recent architecture random-search techniques (Zoph and Le, 2016; Li and Talwalkar, 2019), which take the extreme of applying no human guidance at all. Although as a field we are far from fully understanding the nature of learning and generalization in neural networks, this does not mean that we should proceed blindly.

In this work we define a scaling quantity $\gamma_l$ for each layer $l$ that approximates the average squared singular value of the corresponding diagonal block of the Hessian for layer $l$. This quantity is easy to compute from the (non-central) second moments of the forward-propagated values and the (non-central) second moments of the backward-propagated gradients. We argue that networks that have constant $\gamma_l$ are better conditioned than those that do not, and we analyze how common layer types affect this quantity. We call networks that obey this rule *preconditioned* neural networks, in analogy to preconditioning of linear systems.

As an example of some of the possible applications of our theory, we:

- Propose a *principled* weight initialization scheme that can often provide an improvement over existing schemes;
- Show which common layer types automatically result in well-conditioned networks;
- Show how to improve the conditioning of common structures such as bottlenecked residual blocks by the addition of fixed scaling constants to the network (Detailed in Appendix E).

## 2 NOTATION

We will use the multilayer perceptron (i.e. a classical feed-forward deep neural network) as a running example as it is the simplest non-trivial deep neural network structure. We use ReLU activation functions, and use the following notation for layer $l$ of $L$ (following He et al., 2015):

$$y_l = W_l x_l + b_l,$$

$$x_{l+1} = ReLU(y_l),$$

where $W_l$ is a $n_l^{\text{out}} \times n_l^{\text{in}}$ matrix of weights, $b_l$ is the bias vector, $y_l$ the preactivation vector and $x_l$ is the input activation vector for the layer. The quantities $n_l^{\text{out}}$ and $n_l^{\text{in}}$ are called the fan-out and

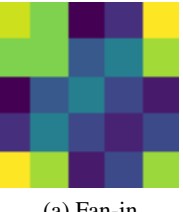 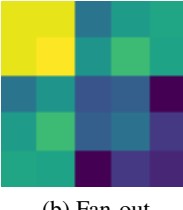 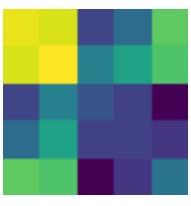 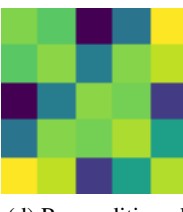

   (a) Fan-in     (b) Fan-out    (c) Arithmetic mean    (d) Preconditioned

Figure 1: Average singular value heat maps for the strided LeNet model, where each square represents a block of the Hessian. The preconditioned network maintains an approximately constant block-diagonal weight. The scale goes from Yellow (larger) through green to blue (smaller).

fan-in of the layer respectively. We also denote the gradient of a quantity with respect to the loss (i.e. the back-propagated gradient) with the prefix $\Delta$. We initially focus on the least-squares loss. Additionally, we assume that each bias vector is initialized with zeros unless otherwise stated.

## 3 CONDITIONING BY BALANCING THE HESSIAN

Our proposed approach focuses on the singular values of the diagonal blocks of the Hessian. In the case of a multilayer perceptron network (MLP) network, each diagonal block corresponds to the weights from a single weight matrix $W_l$ or bias vector $b_l$. This block structure is used by existing approaches such as K-FAC and variants (Martens and Grosse, 2015; Grosse and Martens, 2016; Ba et al., 2017; George et al., 2018), which correct the gradient step using estimates of second-order information. In contrast, our approach modifies the network to improve the Hessian without modifying the step.

Estimates of the magnitude of the singular values $\sigma_i(G_l)$ of the diagonal blocks $G_1,\ldots,G_L$ of the Hessian $G$ provide information about the singular values of the full matrix.

**Proposition 1.** *Let $G_l$ be the lth diagonal block of a real symmetric matrix $G : n \times n$. Then for all $i = 1 \ldots n$:*

$$\sigma_{\min}(G) \leq \sigma_i(G_l),$$
$$\sigma_{\max}(G) \geq \sigma_i(G_l).$$

*We can use this simple bound to provide some insight into the conditioning of the full matrix:*

**Corollary 2.** *Let $S = \{s_1, \ldots, \}$ be the union of the sets of singular values of the diagonal blocks $G_1, \ldots, G_l$ of a real symmetric matrix $G : n \times n$. Then the condition number $\kappa(G) = \sigma_{max}(G)/\sigma_{\min}(G)$ is bounded as:*

$$\kappa(G) \geq \max_{s_i \in S} s_i / \min_{s_i \in S} s_i.$$

In particular, a Hessian matrix with a very large difference between the singular values of each block must be ill-conditioned. This provides strong motivation for balancing the magnitude of the singular values of each diagonal block, the goal of this work. Although ideally, we would like to state the converse, that a matrix with balanced blocks is well conditioned, we can not make such a statement without strong assumptions on the off-diagonal behavior of the matrix.

We use the average squared singular value of each block as a proxy for the full spectrum, as it is particularly easy to estimate in expectation. Although the minimum and maximum for each block would seem like a more natural quantity to work with, we found that any such bounds tend to be too pessimistic to reflect the behavior of the actual singular values.

When using the ReLU activation function, as we consider in this work, a neural network is no longer a smooth function of its inputs, and the Hessian becomes ill-defined at some points in the parameter space. Fortunately, the spectrum is still well-defined at any twice-differentiable point, and this gives a local measure of the curvature. ReLU networks are typically twice-differentiable almost everywhere. We assume this throughout the remainder of this work.

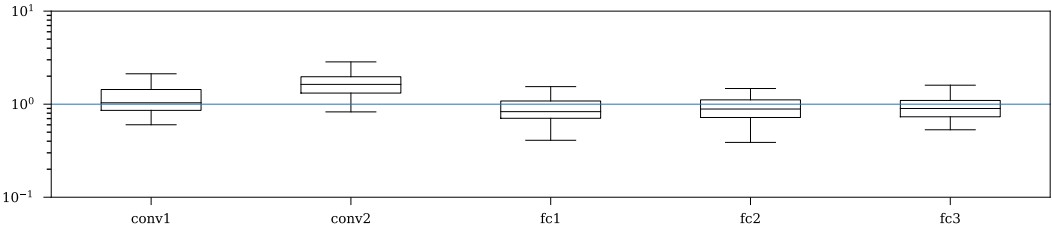

Figure 2: Distributions of the ratio of theoretical scaling to actual for a strided LeNet network

### 3.1 GR SCALING: A MEASURE OF HESSIAN AVERAGE CONDITIONING

Our analysis will proceed with batch-size 1 and a network with $k$ outputs. We consider the network at initialization, where weights are centered, symmetric and i.i.d random variables and biases are set to zero.

ReLU networks have a particularly simple structure for the Hessian with respect to any set of activations, as the network's output is a piecewise-linear function $g$ fed into a final layer consisting of a loss. This structure results in greatly simplified expressions for diagonal blocks of the Hessian with respect to the weights.

We will consider the output of the network as a composition two functions, the current layer $g$, and the remainder of the network $h$. We write this as a function of the weights, i.e. $f(W_l) = h(g(W_l))$. The dependence on the input to the network is implicit in this notation, and the network below layer $l$ does not need to be considered.

Let $R_l = \nabla^2_{y_l} h(y_l)$ be the Hessian of $h$, the remainder of the network after application of layer $l$ (recall $y_l = W_l x_l$). Let $J_l$ be the Jacobian of $y_l$ with respect to $W_l$. The Jacobian has shape $J_l : n_l^{\text{out}} \times \left(n_l^{\text{out}} n_l^{\text{in}}\right)$. Given these quantities, the diagonal block of the Hessian corresponding to $W_l$ is equal to:

$$G_l = J_l^T R_l J_l.$$

The *lth diagonal block of the (Generalized) Gauss-Newton* matrix $G$ (Martens, 2014). We discuss this decomposition further in Appendix A.1. We use the notation $E[X^2]$ for any matrix or vector $X$ to denote the expectation of the element-wise non-central second moment.

**Proposition 3.** *(The GR scaling) Under the assumptions outlined in Section 3.2, the average squared singular value of $G_l$ is equal to the following quantity, which we call the GR scaling for MLP layers:*

$$\textit{(GR scaling)} \quad \gamma_l \doteq n_l^{in} E\left[x_l^2\right]^2 \frac{E[\Delta y_l^2]}{E[y_l^2]}.$$

*We define a "balanced" or "preconditioned" network as one in which $\gamma_l$ is equal for all $l$ (full derivation in Appendix A).*

Balancing this theoretically derived GR scaling quantity in a network will produce an initial optimization problem for which the blocks of the Hessian are expected to be approximately balanced with respect to their average squared singular value.

Due to the large number of approximations needed for this derivation, which we discuss further in the next section, we don't claim that this theoretical approximation is accurate, or that the blocks will be closely matched in practice. Rather, we make the lesser claim that *a network with very disproportionate values of $\gamma_l$ between layers is likely to have convergence difficulties during the early stages of optimization due to Cor. 2.*

To check the quality of our approximation, we computed the ratio of the convolutional version of the GR scaling equation (Equation 1) to the actual $E[(G_l r)^2]$ product for a strided (rather than max-pooled, see Table 1) LeNet model, where we use random input data and a random loss (i.e. for outputs $y$ we use $y^T R y$ for an i.i.d normal matrix $R$), with batch-size 1024, and $32 \times 32$ input images. The results are shown in Figure 2 for 100 sampled setups; there is generally good agreement with the theoretical expectation.

## 3.2 Assumptions

The following strong assumptions are used in the derivation of the GR scaling:

(A1) The input and target values are drawn element-wise i.i.d from a centered symmetric distribution with known variance.

(A2) The Hessian of the remainder of the network above each block, with respect to the output, has Frobenius norm much larger than 1. More concretely, we assume that all but the highest order terms that are polynomial in this norm may be dropped.

(A3) All activations, pre-activations and gradients are independently distributed element-wise. In practice due to the mixing effect of multiplication by random weight matrices, only the magnitudes of these quantities are correlated, and the effect is small for wide networks due to the law of large numbers. Independence assumptions of this kind are common when approximating second-order methods; the block-diagonal variant of K-FAC (Martens and Grosse, 2015) makes similar assumptions for instance.

Assumption A2 is the most problematic of these assumptions, and we make no claim that it holds in practice. However, we are primarily interested in the properties of blocks and their scaling with respect to each other, not their absolute scaling. Assumption A2 results in very simple expressions for the scaling of the blocks without requiring a more complicated analysis of the top of the network. Similar theory can be derived for other assumptions on the output structure, such as the assumption that the target values are much smaller than the outputs of the network.

## 4 Preconditioning also balances weight-to-gradient ratios

We provide further motivation for the utility of preconditioning by comparing it to another simple quantity of interest. Consider at network initialization, the ratio of the (element-wise non-central) second moments of each weight-matrix gradient to the weight matrix itself:

$$\nu_l \doteq \frac{E[\Delta W_l^2]}{E[W_l^2]}.$$

This ratio approximately captures the relative change that a single SGD step with unit step-size on $W_l$ will produce. We call this quantity the weight-to-gradient ratio. When $E[\Delta W_l^2]$ is very small compared to $E[W_l^2]$, the weights will stay close to their initial values for longer than when $E[\Delta W_l^2]$ is large. In contrast, if $E[\Delta W_l^2]$ is very large compared to $E[W_l^2]$, then learning can be expected to be unstable, as the sign of the elements of $W$ may change rapidly between optimization steps. A network with constant $\nu_l$ is also well-behaved under weight-decay, as the ratio of weight-decay second moments to gradient second moments will stay constant throughout the network, keeping the push-pull of gradients and decay constant across the network. Remarkably, the weight-to-gradient ratio $\nu_l$ turns out to be equivalent to the GR scaling for MLP networks:

**Proposition 4.** *(Appendix 8) $\nu_l$ is equal to the GR scaling $\gamma_l$ for i.i.d mean-zero randomly-initialized multilayer perceptron layers under the independence assumptions of Appendix 3.2.*

## 5 Convolutional networks and conditioning multipliers

The concept of GR scaling may be extended to scaled convolutional layers $y_l = \alpha_l Conv_{W_l}(x_l) + b_l$ with scaling factor $\alpha_l$, kernel width $k_l$, batch-size $b$, and output resolution $\rho_l \times \rho_l$ . A straight-forward derivation gives expressions for the convolution weight and biases of:

$$\gamma_l = \alpha_l^4 b n_l^{\text{in}} k_l^2 \rho_l^2 E\left[x_l^2\right]^2 \frac{E[\Delta y_l^2]}{E[y_l^2]}, \qquad \gamma_l^b = \rho_l^2 \frac{E[\Delta y_l^2]}{E[y_l^2]}. \tag{1}$$

This requires an assumption of independence of the values of activations within a channel that is not true in practice, so $\gamma_l$ tends to be further away from empirical estimates for convolutional layers than for non-convolutional layers, although it is still a useful guide. The effect of padding is also ignored here. Sequences of convolutions are well-scaled against each other along as the kernel size remains the same. The scaling of layers involving differing kernel sizes can be corrected using the alpha parameter (Appendix E), and more generally any imbalance between the conditioning of layers

can be fixed by modifying $\alpha_l$ while at the same time changing the initialization of $W_l$ so that the forward variance remains the same as the unmodified version. This adjusts $\gamma_l$ while leaving all other $\gamma$ the same.

## 6 PRECONDITIONING OF NEURAL NETWORKS VIA INITIALIZATION

For ReLU networks with a classical multilayer-perceptron (i.e. non-convolutional, non-residual) structure, we show in this section that initialization using i.i.d mean-zero random variables with (non-central) second moment inversely proportional to the geometric mean of the fans:

$$E[W_l^2] = \frac{c}{\sqrt{n_l^{\text{in}} n_l^{\text{out}}}}, \tag{2}$$

for some fixed constant $c$, results in a constant GR scaling throughout the network.

**Proposition 5.** *Let $W_0 : m \times n$ and $W_1 : p \times m$ be weight matrices satisfying the geometric initialization criteria of Equation 2, and let $b_0, b_1$ be zero-initialized bias parameters. Then consider the following sequence of two layers where $x_0$ and $\Delta y_1$ are i.i.d, mean 0, uncorrelated and symmetrically distributed:*

$$y_0 = W_0 x_0 + b_0, \quad x_1 = ReLU(y_0), \quad y_1 = W_1 x_1 + b_1.$$

*Then $\nu_0 = \nu_1$ and so $\gamma_0 = \gamma_1$.*

*Proof.* Note that the ReLU operation halves both the forward and backward (non-central) second moments, due to our assumptions on the distributions of $x_0$ and $\Delta y_1$. So:

$$E[x_1^2] = \frac{1}{2}E[y_0^2], \quad E[\Delta y_0^2] = \frac{1}{2}E[\Delta x_1^2]. \tag{3}$$

Consider the first weight-gradient ratio, using $E[\Delta W_l^2] = E[x_l^2]E[\Delta y_l^2]$:

$$\frac{E[\Delta W_0^2]}{E[W_0^2]} = \frac{1}{c}E[x_0^2]E[\Delta y_0^2]\sqrt{nm}.$$

Under our assumptions, back-propagation to $\Delta x_1$ results in $E[\Delta x_1^2] = pE[W_1^2]E[\Delta y_1^2]$ , so:

$$E[\Delta y_0^2] = \frac{1}{2}E[\Delta x_1^2] = \frac{1}{2}pE[W_1^2]E[\Delta y_1^2] = \frac{1}{2}p\frac{c}{\sqrt{mp}}E[\Delta y_1^2],$$

So:

$$\frac{E[\Delta W_0^2]}{E[W_0^2]} = \frac{1}{2c}p\frac{c}{\sqrt{mp}}\sqrt{nm}E[x_0^2]E[\Delta y_1^2] = \frac{1}{2}\sqrt{np}E[x_0^2]E[\Delta y_1^2]. \tag{4}$$

Now consider the second weight-gradient ratio:

$$\frac{E[\Delta W_1^2]}{E[W_1^2]} = \frac{1}{c}\sqrt{pm}E[x_1^2]E[\Delta y_1^2].$$

Under our assumptions, applying forward propagation gives $E[y_0^2] = nE[W_0^2]E[x_0^2]$, and so from Equation 3 we have:

$$E[x_1^2] = \frac{1}{2}nE[W_0^2]E[x_0^2] = \frac{1}{2}n\frac{c}{\sqrt{nm}}E[x_0^2],$$

$$\therefore \frac{E[\Delta W_1^2]}{E[W_1^2]} = \frac{1}{2c}\sqrt{pm} \cdot n\frac{c}{\sqrt{nm}}E[x_0^2]E[\Delta y_1^2]$$

$$= \frac{1}{2}\sqrt{np}E[x_0^2]E[\Delta y_1^2],$$

which matches Equation 4, so $\nu_0 = \nu_1$. □

*Remark* 6. This relation also holds for sequences of (potentially) strided convolutions, but only if the same kernel size is used everywhere and circular padding is used. The initialization should be modified to include the kernel size, changing the expression to $c/\left(k_l^2\sqrt{n_l^{\text{in}} n_l^{\text{out}}}\right)$.

### 6.1 TRADITIONAL INITIALIZATION SCHEMES

The most common approaches are the *Kaiming* (He et al., 2015) and *Xavier* (Glorot and Bengio, 2010) initializations. The Kaiming technique for ReLU networks is actually one of two approaches:

$$\text{(fan-in)} \quad \text{Var}[W_l] = \frac{2}{n_l^{\text{in}}} \quad \text{or} \quad \text{(fan-out)} \quad \text{Var}[W_l] = \frac{2}{n_l^{\text{out}}}. \tag{5}$$

For the feed-forward network above, assuming random activations, the forward-activation variance will remain constant in expectation throughout the network if fan-in initialization of weights (LeCun et al., 2012) is used, whereas the fan-out variant maintains a constant variance of the back-propagated signal. The constant factor 2 in the above expressions corrects for the variance-reducing effect of the ReLU activation. Although popularized by He et al. (2015), similar scaling was in use in early neural network models that used tanh activation functions (Bottou, 1988).

These two principles are clearly in conflict; unless $n_l^{\text{in}} = n_l^{\text{out}}$, either the forward variance or backward variance will become non-constant, or as more commonly expressed, either *explode* or *vanish*. No *prima facie* reason for preferring one initialization over the other is provided. Unfortunately, there is some confusion in the literature as many works reference using Kaiming initialization without specifying if the fan-in or fan-out variant is used.

The Xavier initialization (Glorot and Bengio, 2010) is the closest to our proposed approach. They balance these conflicting objectives using the arithmetic mean:

$$\text{Var}[W_l] = \frac{4}{n_l^{\text{in}} + n_l^{\text{out}}}, \tag{6}$$

to "... approximately satisfy our objectives of maintaining activation variances and back-propagated gradients variance as one moves up or down the network". This approach to balancing is essentially heuristic, in contrast to the geometric mean approach that our theory directly guides us to.

### 6.2 GEOMETRIC INITIALIZATION BALANCES BIASES

We can use the same proof technique to compute the GR scaling for the bias parameters in a network. Our update equations change to include the bias term: $y_l = W_l x_l + b_l$, with $b_l$ assumed to be initialized at zero. We show in Appendix D that:

$$\gamma_l^b = \frac{E[\Delta y_l^2]}{E[y_l^2]}.$$

It is easy to show using the techniques of Section 6 that the biases of consecutive layers have equal GR scaling as long as geometric initialization is used. However, unlike in the case of weights, we have less flexibility in the choice of the numerator. Instead of allowing all weights to be scaled by $c$ for any positive $c$, we require that $c = 2$, so that:

$$E[W_l^2] = \frac{2}{\sqrt{n_l^{\text{in}} n_l^{\text{out}}}}. \tag{7}$$

**Proposition 7.** *(Appendix D) Consider the setup of Proposition 5. As long as the weights are initialized following Equation 7 with $c = 2$ and the biases are initialized to 0, we have that $\gamma_0^b = \gamma_1^b$.*

### 6.3 NETWORK INPUT SCALING BALANCES WEIGHTS AGAINST BIASES

It is traditional to normalize a dataset before applying a neural network so that the input vector has mean 0 and variance 1 in expectation. This principle is rarely quested in modern neural networks, even though there is no longer a good justification for its use in modern ReLU based networks. In contrast, our theory provides direct guidance for the choice of input scaling. We show that the (non-central) second moment of the input affects the GR scaling of bias and weight parameters differently and that they can be balanced by careful choice of the initialization.

Consider the GR scaling values for the bias and weight parameters in the first layer of a ReLU-based multilayer perceptron network, as considered in previous sections. We assume the data is already

Table 1: Scaling of common layers

| Method | Maintains Scaling | Notes |
|---|---|---|
| Linear layer | ✓ | Will not be well-conditioned against other layers unless geometric initialization is used |
| (Strided) convolution | ✓ | As above, but only if all kernel sizes are the same |
| Skip connections | ✗ | Operations in residual blocks will be scaled correctly against each other, but not against non-residual operations |
| Average pooling | ✓ | |
| Max pooling | ✗ | |
| Dropout | ✓ | |
| ReLU/LeakyReLU | ✓ | Any positively-homogenous function with degree 1 |
| Sigmoid | ✗ | |
| Tanhh | ✗ | Maintains scaling if entirely within the linear regime |

centered. Then the scaling factors for the weight and bias layers are:

$$\gamma_0 = n_0^{\text{in}} k_0^2 \rho_0^2 E\left[x_0^2\right]^2 \frac{E[\Delta y_0^2]}{E[y_0^2]}, \qquad \gamma_{0b} = \rho_0^2 \frac{E[\Delta y_0^2]}{E[y_0^2]}.$$

We can cancel terms to find the value of $E\left[x_0^2\right]$ that makes these two quantities equal:

$$E\left[x_0^2\right] = \frac{1}{\sqrt{n_0^{\text{in}} k_0^2}}.$$

In common computer vision architectures, the input planes are the 3 color channels, and the kernel size is $k = 3$, giving $E\left[x_0^2\right] \approx 0.2$. Using the traditional variance-one normalization will result in the effective learning rate for the bias terms being lower than that of the weight terms. This will result in potentially slower learning of the bias terms than for the input scaling we propose.

### 6.4 OUTPUT (NON-CENTRAL) SECOND MOMENTS

A neural network's behavior is also very sensitive to the (non-central) second moment of the outputs. For a convolutional network without pooling layers (but potentially with strided dimensionality reduction), if geometric-mean initialization is used the activation (non-central) second moments are given by:

$$E[x_{l+1}^2] = \frac{1}{2} k^2 n_l^{in} E[W_l^2] E[x_l^2] = \sqrt{\frac{n_l^{in}}{n_l^{out}}} E[x_l^2].$$

The application of a sequence of these layers gives a telescoping product:

$$E[x_{L+1}^2] = \left(\prod_{l=0}^{L} \sqrt{\frac{n_l^{in}}{n_l^{out}}}\right) E[x_0^2] = \sqrt{\frac{n_0^{in}}{n_L^{out}}} E[x_0^2].$$

We potentially have independent control over this (non-central) second moment at initialization, as we can insert a fixed scalar multiplication factor at the end of the network that modifies it. This may be necessary when adapting a network architecture that was designed and tested under a different initialization scheme, as the success of the architecture may be partially due to the output scaling that happens to be produced by that original initialization. We are not aware of any existing theory guiding the choice of output variance at initialization for the case of log-softmax losses, where it has a non-trivial effect on the back-propagated signals, although output variances of 0.01 to 0.1 appear to work well. The output variance should **always** be checked and potentially corrected when switching initialization schemes.

## 7 DESIGNING WELL-CONDITIONED NEURAL NETWORKS

Consider a network where $\gamma_l$ is constant throughout. We may add a layer between any two existing layers without affecting this conditioning, as long as the new layer maintains the activation-gradient (non-central) second-moment product:

$$E[\Delta x_{l+1}^2] E[x_{l+1}^2] = E[\Delta x_l^2] E[x_l^2],$$

Table 2: Comparison on 26 LIBSVM repository datasets

| Method | Average Normalized loss ($\pm 0.01$) | Worst in # | Best in # |
|---|---|---|---|
| Arithmetic mean | 0.90 | 14 | 3 |
| Fan-in | 0.84 | 3 | 5 |
| Fan-out | 0.88 | 9 | **12** |
| Geometric mean | **0.81** | **0** | 6 |

and dimensionality; this follows from Equation 1. For instance, adding a simple scaling layer of the form $x_{l+1} = 2x_l$ doubles the (non-central) second moment during the forward pass and doubles the backward (non-central) second moment during back-propagation, which maintains this product:

$$E[\Delta x_{l+1}^2]E[x_{l+1}^2] = \frac{1}{2}E[\Delta x_l^2] \cdot 2E[x_l^2].$$

When spatial dimensionality changes between layers we can see that the GR scaling is no longer maintained just by balancing this product, as $\gamma$ depends directly on the square of the spatial dimension. Instead, a pooling operation that changes the forward and backward signals in a way that counteracts the change in spatial dimension is needed. The use of stride-2 convolutions, as well as average pooling, results in the correct scaling, but other common types of spatial reduction generally do not. Table 1 lists operations that preserve scaling when inserted into an existing preconditioned network. Operations such as linear layers preserve the scaling of existing layers but are only themselves well-scaled if they are initialized correctly. For an architecture such as ResNet-50 that uses operations that break scaling, $\alpha_l$ values can be introduced to correct scaling. In a ResNet-50, residual connections, max-pooling, and varying kernel sizes need to be corrected for (we describe this procedure in Appendix E).

It is particularly interesting to note that the evolution in state-of-the-art architectures corresponds closely to a move from poorly scaled building blocks to well-scaled ones. Early shallow architectures like LeNet-5 used tanh nonlinearities, which were replaced by the (well-scaled) ReLU, used for instance in the seminal AlexNet architecture (Krizhevsky et al., 2012). AlexNet and the latter VGG architectures made heavy use of max-pooling and reshaping before the final layers, both operations which have been replaced in modern fully-convolutional architectures with (well-scaled) striding and average-pooling respectively. The use of large kernel sizes is also in decline. The AlexNet architecture used kernel sizes of 11, 5 and 3, whereas modern ResNet (He et al., 2016) architectures only use 7, 3 and 1. Furthermore, recent research has shown that replacing the single 7x7 convolution used with a sequence of three 3x3 convolutions improves performance (He et al., 2018).

## 8 EXPERIMENTAL RESULTS

We considered a selection of dense and moderate-sparsity multi-class classification datasets from the LibSVM repository, 26 in total. The same model was used for all datasets, a non-convolutional ReLU network with 3 weight layers total. The inner two layer widths were fixed at 384 and 64 nodes respectively. These numbers were chosen to result in a larger gap between the optimization methods, less difference could be expected if a more typical $2\times$ gap was used. Our results are otherwise generally robust to the choice of layer widths.

For every dataset, learning rate and initialization combination we ran 10 seeds and picked the median loss after 5 epochs as the focus of our study (The largest differences can be expected early in training). Learning rates in the range $2^1$ to $2^{-12}$ (in powers of 2) were checked for each dataset and initialization combination, with the best learning rate chosen in each case based off of the median of the 10 seeds. Training loss was used as the basis of our comparison as we care primarily about convergence rate, and are comparing identical network architectures. Some additional details concerning the experimental setup and which datasets were used is available in the Appendix.

Table 1 shows that geometric initialization is the most consistent of the initialization approaches considered. It has the lowest loss, after normalizing each dataset, and it is never the worst of the 4 methods on any dataset. Interestingly, the fan-out method is most often the best method, but consideration of the per-dataset plots (Appendix F) shows that it often completely fails to learn for some problems, which pulls up its average loss and results in it being the worst for 9 datasets.

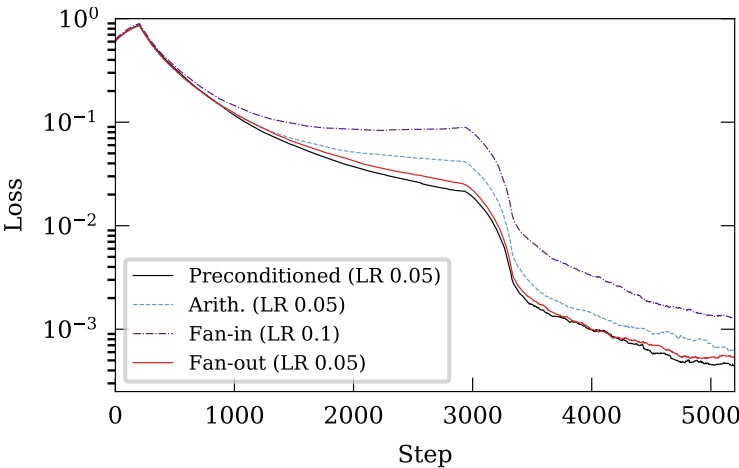

Figure 3: CIFAR-10 training loss for a strided AlexNet architecture. An average of 10 seeds is shown for each initialization, where for each seed a sliding window of minibatch training loss over 400 steps was used.

## 8.1 CONVOLUTIONAL EXPERIMENTS

Testing an initialization method on modern computer vision problems is problematic due to the heavy architecture search, both automated and manual, that is behind the current best methods. This search will fit the architecture to the initialization method, in a sense, so any other initialization is at a disadvantage compared to the one used during architecture search. This is further complicated by the prevalence of BatchNorm which is not handled in our theory. Instead, to provide a clear comparison we use an older network with a large variability in kernel sizes, the AlexNet architecture. This architecture has a large variety of filter sizes (11, 5, 3, linear), which according to our theory will affect the conditioning adversely, and which should highlight the differences between the methods. We found that a network with consistent kernel sizes through-out showed only negligible differences between the initialization methods. The network was modified to replace max-pooling with striding as max-pooling is not well-scaled by our theory (further details in Appendix F).

Following Section 6.4, we normalize the output of the network at initialization by running a single batch through the network and adding a fixed scaling factor to the network to produce output standard deviation 0.05. For our preconditioned variant, we added alpha correction factors following Section 5 in conjunction with geometric initialization, and compared this against other common initialization methods. We tested on CIFAR-10 following the standard practice as closely as possible, as detailed in Appendix F. We performed a geometric learning rate sweep over a power-of-two grid. Results are shown in Figure 3 for an average of 10 seeds for each initialization. Preconditioning improves training loss over all other initialization schemes tested, although only by a small margin.

## 9 CONCLUSION

Although not a panacea, by using the scaling principle we have introduced, neural networks can be designed with a reasonable expectation that they will be optimizable by stochastic gradient methods, minimizing the amount of guess-and-check neural network design. As a consequence of our scaling principle, we have derived an initialization scheme that automatically preconditions common network architectures. Most developments in neural network theory attempt to explain the success of existing techniques post-hoc. Instead, we show the power of the scaling law approach by deriving a new initialization technique from theory directly.

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

## A  GR SCALING DERIVATION

Our quantity of interest is the average squared singular value of $G_l$, which is simply equal to the (element-wise) non-central second moment of the product of $G$ with a i.i.d normal random vector $r$:

$$E[(G_l r)^2] = E[\left(J_l^T R_l J_l r\right)^2].$$

Recall that our notation $E[X^2]$ refers to the element-wise non-central second moment of the vector. Through-out the remainder of this work we use non-central moments unless otherwise stated. To compute the second moment of the elements of $G_l r$, we can calculate the second moment of matrix-random-vector products against $J_l$, $R_l$ and $J_l^T$ separately since $R$ is uncorrelated with $J_l$, and the back-propagated gradient $\Delta y_l$ is uncorrelated with $y_l$ (Assumption A3).

JACOBIAN PRODUCTS $J_l$ AND $J_l^T$

Note that each row of $J_l$ has $n_l^{\text{in}}$ non-zero elements, each containing a value from $x_l$. This structure can be written as a block matrix,

$$J_l = \begin{bmatrix} x_l & 0 & 0 \\ 0 & x_l & 0 \\ 0 & 0 & \ddots \end{bmatrix}, \tag{8}$$

Where each $x_l$ is a $1 \times n_l^{\text{in}}$ row vector. This can also be written as a Kronecker product with an identity matrix as $I_{n_l^{\text{out}}} \otimes x_l$. The value $x_l$ is i.i.d random at the bottom layer of the network (Assumption A1). For layers further up, the multiplication by a random weight matrix from the previous layer ensures that the entries of $x_l$ are identically distributed (Assumption A3). So we have:

$$E\left[(J_l r)^2\right] = n_l^{\text{in}} E[r^2] E[x_l^2] = n_l^{\text{in}} E[x_l^2]. \tag{9}$$

Note that we didn't assume that the input $x_l$ is mean zero, so $Var[x_l] \neq E[x_l^2]$. This is needed as often the input to a layer is the output from a ReLU operation, which will not be mean zero.

For the transposed case, we have a single entry per column, so when multiplying by an i.i.d random vector $u$ we have:

$$E\left[\left(J_l^T u\right)^2\right] = E[u^2] E[x_l^2]. \tag{10}$$

UPPER HESSIAN $R_l$ PRODUCT

Instead of using $R_l u$, for a random $u$, we will instead compute it for $u = y_l / E[y_l^2]$, it will have the same expectation since both $J_l r$ and $y_l$ are uncorrelated with $R_l$ (Assumption A3). The piecewise linear structure of the network above $y_l$ with respect to the $y_l$ makes the structure of $R_l$ particularly simple. It is a least-squares problem $g(y_l) = \frac{1}{2} \|\Phi y_l - t\|^2$ for some $\Phi$ that is the linearization of the remainder of the network. The gradient is $\Delta y = \Phi^T (\Phi y - t)$ and the Hessian is simply $R = \Phi^T \Phi$. So we have that

$$\begin{aligned} E\left[\Delta y_l^2\right] &= E\left[\frac{1}{n_l^{\text{out}}} \left\|\Phi^T (\Phi y - t)\right\|^2\right] \\ &= E\left[\frac{1}{n_l^{\text{out}}} \left\|\Phi^T \Phi y\right\|^2\right] + E\left[\frac{1}{n_l^{\text{out}}} \left\|\Phi^T t\right\|^2\right] \quad \text{(Uncorr. A1)} \\ &\approx E\left[\frac{1}{n_l^{\text{out}}} \left\|\Phi^T \Phi y\right\|^2\right] = E\left[(R_l y_l)^2\right]. \quad \text{(Assumption A2)} \end{aligned}$$

Applying this gives:

$$E\left(R_l u\right)^2 = E[u^2] E[(R_l y_l)^2]/E[y_l^2] = E[u^2] E[\Delta y_l^2]/E[y_l^2]. \tag{11}$$

COMBINING

To compute $E[(G_l r)^2] = E[(J_l^T R_l J_l r)^2]$ we then combine the simplifications from Equations 9, 10, and 11 to give:

$$E[(G_l r)^2] = n_l^{\text{in}} E\left[x_l^2\right]^2 \frac{E[\Delta y_l^2]}{E[y_l^2]}.$$

## A.1 THE GAUSS-NEWTON MATRIX

Standard ReLU classification and regression networks have a particularly simple structure for the Hessian with respect to the input, as the network's output is a piecewise-linear function $g$ feed into a final layer consisting of a convex log-softmax operation, or a least-squares loss. This structure results in the Hessian with respect to the input being equivalent to its *Gauss-Newton* approximation. The Gauss-Newton matrix can be written in a factored form, which is used in the analysis we perform in this work. We emphasize that this is just used as a convenience when working with diagonal blocks, the GN representation is not an approximation in this case.

The (Generalized) Gauss-Newton matrix $G$ is a positive semi-definite approximation of the Hessian of a non-convex function $f$, given by factoring $f$ into the composition of two functions $f(x) = h(g(x))$ where $h$ is convex, and $g$ is approximated by its Jacobian matrix $J$ at $x$, for the purpose of computing $G$:

$$G = J^T \left(\nabla^2 h(g(x))\right) J.$$

The GN matrix also has close ties to the Fisher information matrix (Martens, 2014), providing another justification for its use.

Surprisingly, the Gauss-Newton decomposition can be used to compute diagonal blocks of the Hessian with respect to the weights $W_l$ as well as the inputs (Martens, 2014). To see this, note that for any activation $y_l$, the layers above may be treated in a combined fashion as the $h$ in a $f(W_l) = h(g(W_l))$ decomposition of the network structure, as they are the composition of a (locally) linear function and a convex function and thus convex. In this decomposition $g(W_l) = W_l x_l + b_l$ is a function of $W_l$ with $x_l$ fixed, and as this is linear in $W_l$, the Gauss-Newton approximation to the block is thus not an approximation.

## B  FORWARD AND BACKWARD SECOND MOMENTS

We make heavy use of the equations for forward propagation and backward propagation of second moments, under the assumption that the weights are uncorrelated to the activations or gradients. For a convolution

$$y = C_W(x)$$

with input channels $n^{\text{in}}$, output channels $n^{\text{out}}$, and square $k \times k$ kernels, these formulas are (recall our notation for the second moments is element-wise for vectors and matrices):

$$E[y^2] = n^{\text{in}} k^2 E[W^2] E[x^2],$$
$$E[\Delta x^2] = n^{\text{out}} k^2 E[W^2] E[\Delta y^2].$$

## C  THE WEIGHT GRADIENT RATIO IS EQUAL TO GR SCALING FOR MLP MODELS

**Proposition 8.** *The weight-gradient ratio $\nu_l$ is equal to the GR scaling $\gamma_l$ for i.i.d mean-zero randomly-initialized multilayer perceptron layers under the independence assumptions of Appendix 3.2.*

*Proof.* To see the equivalence, note that under the zero-bias initialization, we have from $y_l = W_l x_l$ that:

$$E[y_l^2] = n_l^{\text{in}} E[W_l^2] E[x_l^2], \tag{12}$$

and so:

$$E[W_l^2] = \frac{E[y_l^2]}{n_l^{in} E[x_l^2]}.$$

The gradient of the weights is given by $\Delta W_{ij} = \Delta y_{li} x_{lj}$ and so its second moment is:

$$E[\Delta W_l^2] = E[x_l^2] E[\Delta y_l^2]. \tag{13}$$

Combining these quantities gives:

$$\nu_l = \frac{E[\Delta W_l^2]}{E[W_l^2]} = n_l^{in} E\left[x_l^2\right]^2 \frac{E[\Delta y_l^2]}{E[y_l^2]}.$$

$$\square$$

## D   BIAS SCALING

We consider the case of a convolutional neural network with spatial resolution $\rho \times \rho$ for greater generality. Consider the Jacobian of $y_l$ with respect to the bias. It has shape $J_l^b : (n_l^{out} \rho_l^2) \times (n_l^{out})$. Each row corresponds to a $y_l$ output, and each column a bias weight. As before, we will approximate the product of $G$ with a random i.i.d unit variance vector $r$:

$$G_l^b r = J_l^{bT} R_l J_l^b r,$$

The structure of $J_l^b$ is that each block of $\rho^2$ rows has the same set of 1s in the same column. Only a single 1 per row. It follows that:

$$E\left[\left(J_l^b r\right)^2\right] = 1.$$

The calculation of the product of $R_l$ with $J_l^b r$ is approximated in the same way as in the weight scaling calculation. For the $J^{bT}$ product, note that there is an additional $\rho^2$ as each column has $\rho^2$ non-zero entries, each equal to 1. Combining these three quantities gives:

$$\gamma_l^b = \rho^2 \frac{E[\Delta y_l^2]}{E[y_l^2]}.$$

**Proposition 9.** *Consider the setup of Proposition 5, with the addition of biases:*

$$y_0 = W_0 x_0 + b_0,$$

$$x_1 = ReLU(y_0),$$

$$y_1 = W_1 + b_1.$$

*As long as the weights are initialized following Equation 7 and the biases are initialized to 0, we have that*

$$\gamma_0^b = \gamma_1^b.$$

We will include $c = 2$ as a variable as it clarifies it's relation to other quantities. We reuse some calculations from Proposition 5. Namely that:

$$E[y_0^2] = c\sqrt{\frac{n}{m}} E[x_0^2],$$

$$E[\Delta y_0^2] = \frac{1}{2} c\sqrt{\frac{p}{m}} E[\Delta y_1^2].$$

Plugging these into the definition of $\gamma_0^b$:

$$\gamma_0^b = \frac{E[\Delta y_0^2]}{E[y_0^2]} = \frac{\frac{1}{2} c\sqrt{\frac{p}{m}} E[\Delta y_1^2]}{c\sqrt{\frac{n}{m}} E[x_0^2]} = \frac{\sqrt{p} E[\Delta y_1^2]}{2\sqrt{n} E[x_0^2]}.$$

For $\gamma_1^b$, we require the additional quantity:

$$E[y_1^2] = mE[x_1^2]E\left[W_1^2\right]$$
$$= m\left(\frac{1}{2}c\sqrt{\frac{n}{m}}E[x_0^2]\right)\left(\frac{c}{\sqrt{mp}}\right)$$
$$= \frac{c^2}{2}\sqrt{\frac{n}{p}}E[x_0^2].$$

Again plugging this in:

$$\gamma_1^b = \frac{E[\Delta y_1^2]}{E[y_1^2]}$$
$$= \frac{E[\Delta y_1^2]}{\frac{c^2}{2}\sqrt{\frac{n}{p}}E[x_0^2]}$$
$$= \frac{\sqrt{p}E[\Delta y_1^2]}{\frac{c^2}{2}\sqrt{n}E[x_0^2]}.$$

So comparing these expressions for $\gamma_0^b$ and $\gamma_1^b$, we see that $\gamma_0^b = \gamma_1^b$ if and only if $c = 2$.

## E    CONDITIONING OF RESNETS WITHOUT NORMALIZATION LAYERS

There has been significant recent interest in training residual networks without the use of batch-normalization or other normalization layers (Zhang et al., 2019). In this section, we explore the modifications that are necessary to a network for this to be possible and show how to apply our preconditioning principle to these networks.

The building block of a ResNet model is the residual block:

$$z_{l+1} = \text{ReLU}\left(F(z_l) + z_l\right),$$

where $F$ is a composition of layers. Unlike classical feedforward architectures, the pass-through connection results in an exponential increase in the variance of the activations in the network as the depth increases. A side effect of this is the output of the network becomes exponentially more sensitive to the input of the network as depth increases, a property characterized by the Lipschitz constant of the network (Hanin, 2018).

This exponential dependence can be reduced by the introduction of scaling constants $s_l$ to each block:

$$z_{l+1} = \text{ReLU}\left(s_l F(z_l) + z_l\right).$$

The introduction of these constants requires a modification of the block structure to ensure constant conditioning between blocks. A standard bottleneck block, as used in the ResNet-50 architecture, has the following form:

$$y_0 = C_0(x_0),$$
$$x_1 = \text{ReLU}(y_0),$$
$$y_1 = C_1(x_1),$$
$$x_2 = \text{ReLU}(y_1),$$
$$y_2 = C_2(x_2),$$
$$x_3 = \text{ReLU}(y_2 + x_0).$$

In this notation, $C_0$ is a $1 \times 1$ convolution that reduces the number of channels 4 fold, $C_1$ is a $3 \times 3$ convolution with equal input and output channels, and $C_2$ is a $1 \times 1$ convolution at increases the number of channels back up 4 fold to the original input count.

If we introduce a scaling factor $s_l$ to each block $l$, then we must also add conditioning multipliers $\beta_l$ to each convolution to change their GR scaling, as we described in Section 5. The correct scaling

constant depends on the scaling constant of the previous block. A simple calculation gives the equation:

$$\beta_l^2 = \beta_{l-1}^2 \frac{1 + s_l^2}{1 + s_{l-1}^2}.$$

The initial $\beta_0$ and $s_0$ may be chosen arbitrarily. If a flat $s_l = s$ is used for all $l$, then we may use $\beta_l = 1$. The block structure including the $\beta_l$ factors is:

$$y_0 = \frac{1}{\beta} C_0(x_0),$$
$$x_1 = \text{ReLU}(y_0),$$
$$y_1 = \frac{1}{\sqrt{3}\beta} C_1(x_1),$$
$$x_2 = \text{ReLU}(y_1),$$
$$y_2 = \frac{1}{\beta} C_2(x_2),$$
$$x_3 = \text{ReLU}\left(sy_2 + x_0\right)$$

The weights of each convolution must then be initialized with the standard deviation modified such that the combined convolution-scaling operation gives the same output variance as would be given if the geometric-mean initialization scheme is used without extra scaling constants. For instance, the initialization of the $C_0$ convolution must have standard deviation scaled down by dividing by $\frac{1}{\beta}$ so that the multiplication by $\frac{1}{\beta}$ during the forward pass results in the correct forward variance. The $1/\sqrt{3}$ factor is an $\alpha$ correction that corrects for change in kernel shape for the middle convolution. The variance at initialization must be scaled to correct for the $\alpha$ factor also.

### E.1 Correction for mixed residual and non-residual blocks

Since the initial convolution in a ResNet-50 model is also not within a residual block, it's GR scaling is different from the convolutions within residual blocks. Consider the composition of a non-residual followed by a residual block, without max-pooling or ReLUs for simplicity of exposition:

$$y_0 = \alpha C_0(x_0), \quad x_1 = y_0,$$
$$y_1 = s_1 C_1(x_1), \quad z_1 = y_1 + x_1.$$

Without loss of generality, we assume that $E\left[x_0^2\right] = 1$, and assume a single channel input and output.

Our goal is to find a constant $\alpha$, so that $\gamma_0 = \gamma_1$. Recall that when using $\alpha$ scaling factors we must initialize $C_0$ so that the variance of $y_0$ is independent of the choice of $\alpha$. Our scaling factor will also depend on the kernel sizes used in the two convolutions, so we must include those in the calculations.

From Equation 1, the GR scaling for $C_0$ is

$$\gamma_0 = \alpha^4 n_l^{\text{in}} k_0^2 E\left[x_0^2\right]^2 \frac{E[\Delta y_0^2]}{E[y_0^2]}$$
$$= \alpha^4 k_0^2 E[\Delta y_0^2].$$

Note that $E[\Delta y_0^2] = \left(1 + s_1^2\right) E[\Delta z_1^2]$ so:

$$\gamma_0 = \left(1 + s_1^2\right) \alpha^4 k_0^2 E[\Delta z_1^2],$$

For the residual convolution, we need to use a modification of the standard GR equation due to the residual branch. The derivation of $\gamma$ for non-residual convolutions assumes that the remainder of the network above the convolution responds linearly (locally) with the scaling of the convolution, but here due to the residual connection, this is no longer the case. For instance, if the weight were scaled to zero, the output of the network would not also become zero (recall our assumption of

zero-initialization for bias terms). This can be avoided by noting that the ratio $E[\Delta y_1^2]/E[y_1^2]$ in the GR scaling may be computed further up the network, as long as any scaling in between is corrected for. In particular, we may compute this ratio at the point after the residual addition, as long as we include the factor $s_1^4$ to account for this. So we in fact have:

$$\gamma_1 = s_1^4 n_l^{\text{in}} k_1^2 E\left[x_1^2\right]^2 \frac{E[\Delta z_1^2]}{E[z_1^2]}$$

$$= s_1^4 k_1^2 \frac{E[\Delta z_1^2]}{1 + s_1^2}.$$

We now equate $\gamma_0 = \gamma_1$:

$$s_1^4 k_1^2 \frac{E[\Delta z_1^2]}{1 + s_1^2} = \left(1 + s_1^2\right) \alpha^4 k_0^2 E[\Delta z_1^2],$$

$$\frac{k_1^2}{k_0^2} \cdot \frac{s_1^4}{\left(1 + s_1^2\right)^2} = \alpha^4.$$

Therefore to ensure that $\gamma_0 = \gamma_1$ we need:

$$\alpha^2 = \frac{k_1}{k_0} \cdot \frac{s_1^2}{\left(1 + s_1^2\right)}.$$

### FINAL LAYER

A similar calculation applies when the residual block is before the non-residual convolution, as in the last layer linear in the ResNet network, giving a scaling factor for the linear layer (effective kernel size 1) of:

$$\alpha^2 = \frac{s_{L-1}^2}{\left(1 + s_{L-1}^2\right)} k_{L-1}.$$

## F   FULL EXPERIMENTAL RESULTS

### DETAILS OF LIBSVM DATASET INPUT/OUTPUT SCALING

To prevent the results from being skewed by the number of classes and the number of inputs affecting the output variance, the logit output of the network was scaled to have standard deviation 0.05 after the first minibatch evaluation for every method, with the scaling constant fixed thereafter. LayerNorm was used on the input to whiten the data. Weight decay of 0.00001 was used for every dataset. To aggregate the losses across datasets we divided by the worst loss across the initializations before averaging.

### LIBSVM PLOTS

Figure 4 shows the interquartile range (25%, 50% and 75% quantiles) of the best learning rate for each case.

### CIFAR10

Following standard practice, training used random augmentations consisting of horizontal flips and random crops to 32x32, as well as normalization to the interval [-1,+1]. We used SGD with momentum 0.9, a learning rate schedule of decreases at 150 and 225 epochs, and no weight decay.

The network architecture is the following sequence, with circular "equal" padding used and ReLU nonlinearities after each convolution:

1. 11x11 stride-1 convolution with 3 input and 64 output channels,
2. 5x5 stride-2 convolution with 64 input and 192 output channels,
3. 3x3 stride-2 convolution with 192 input and 384 output channels,

4. 3x3 convolution with 384 input and 256 output channels,
5. 3x3 convolution with 256 input and 256 output channels,
6. Average pooling to 1x1 followed by reshaping to non-spatial,
7. Linear layer with 256 input and 4096 output features,
8. Linear layer with 4096 input and 4096 output features,
9. Linear layer with 4096 input and 10 output features,

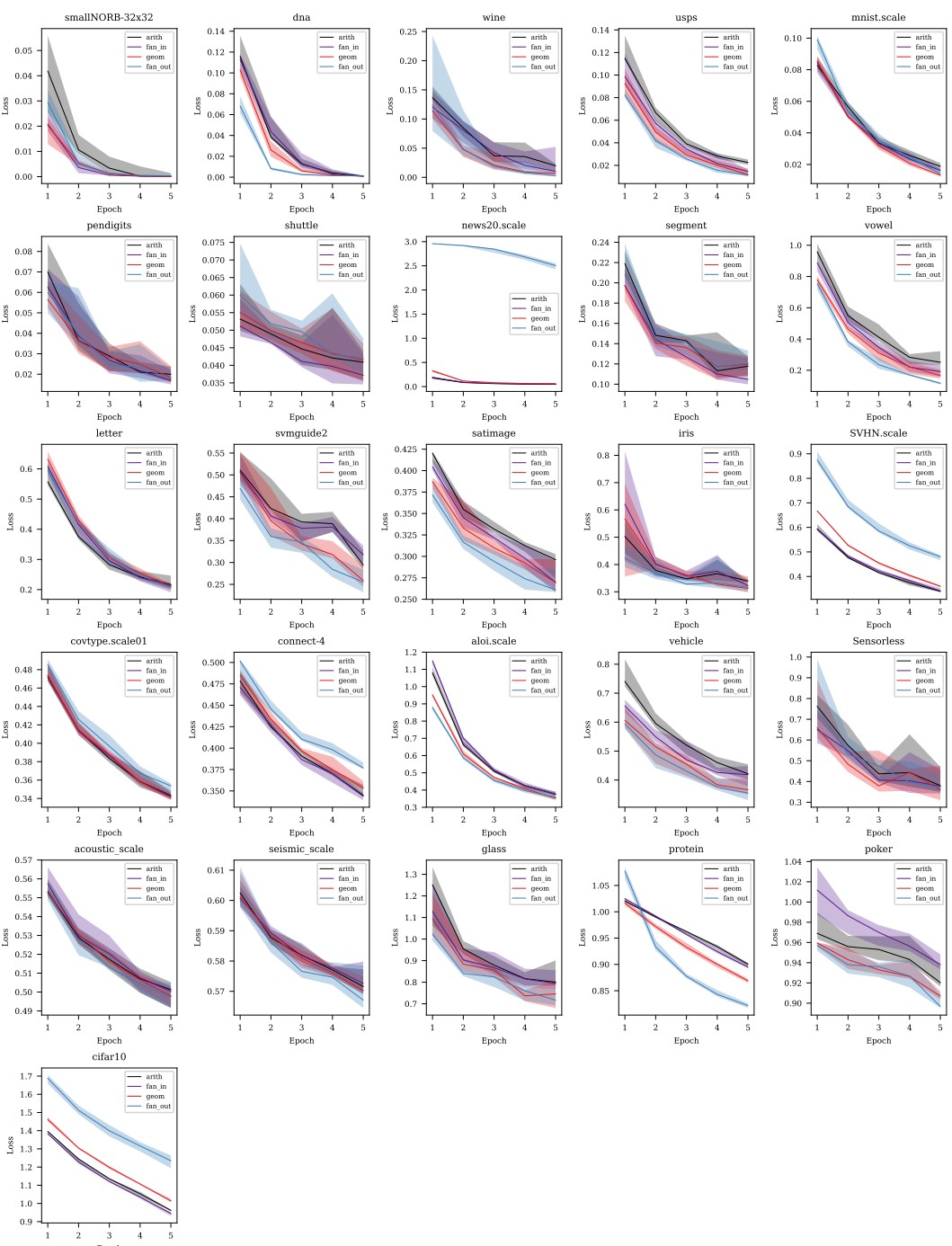

Figure 4: Full results for the 26 datasets

