# OpenReview forum: "Scaling Laws for the Principled Design, Initialization, and Preconditioning of ReLU Networks"
_ICLR.cc/2020/Conference — Reject_

### Official Review · AnonReviewer3 · 2019-10-22
**Official Blind Review #3**

**Rating:** 1

**Review:**

The paper proposes a set of rules for the design and initialization of well-conditioned neural networks by naturally balancing the diagonal blocks of the Hessian at the start of training. Overall, the paper is well written and clear in comparison and explanation. However, the reviewer is concerned with the following questions:
Assumption A2 does not make sense in the context. In particular, it is not praised to assume it only for the convenience of computation without giving any example when the assumption would hold. Also the assumptions are vague and hard to understand, it is better to have concrete mathematical formulation after text description.
Are the experiment results sensitive to the choice of different models with different width and layers or different batch sizes? Does it have a strong improvement than random initialization? It’s less clear the necessity of guaranteeing well-conditioned at initialization since during the training procedure, the condition number is harder to control.


**Experience Assessment:**

I have read many papers in this area.

**Review Assessment: Checking Correctness Of Derivations And Theory:**

I assessed the sensibility of the derivations and theory.

**Review Assessment: Checking Correctness Of Experiments:**

I assessed the sensibility of the experiments.

**Review Assessment: Thoroughness In Paper Reading:**

I read the paper at least twice and used my best judgement in assessing the paper.

---

> ### Author Response · Authors · 2019-11-05
> **Reply**
>
> Thank you for taking the time to review our work. We provide the following replies:
>
> -Our results are quite robust, note that the input and output dimensionality varies across the 27 datasets we compare on, resulting in different layer widths at the beginning and end of the network for every dataset. In addition, the AlexNet we test on has a very different architecture from the other experiments.
>
> - Thank you for the feedback about assumption A2. Some assumption is needed to control the behavior at the top of the network. Note that we are talking about the network at initialization, a situation where the assumption is mild.
>
> Given the extensive theory that results from this simple assumption we believe it is interesting in any case. In general, developing assumptions that result in fruitful and simple analysis is one of the primary goals of theory, and should not be seen as a weakness.
> We are looking to provide a theory that handles more general cases in future work.
>
>
> We hope that Reviewer #3 reconsiders their assessment given the context from other reviewers and our replies above.

---

### Official Review · AnonReviewer2 · 2019-10-23
**Official Blind Review #2**

**Rating:** 3

**Review:**

The authors propose a new initialization scheme for training neural networks. The initialization considers fan-in and fan-out, to regularize the range of singular values of the Hessian matrix, under several assumptions.

The proposed approach gives important insights for the problem of weight initialization in neural networks. Overall, the method makes sense. However, I have several concerns:

- The authors do not consider more recent neural network designs such as normalization layers, skip connections, etc. It would be great if the authors could discuss how these layers would change the derivation of the initialization method. Also, preliminary experimental results using these layers are needed. Additionally, to me, normalization layers [Huang et al. Decorrelated Batch Normalization. CVPR 2018] implicitly precondition the Hessian matrix as well. It would be great if the authors also compare their approach to [Huang et al. 2018].

- The authors compared to other initialization schemes such as [He et al., 2015] and [ Glorot and Bengio 2010]. But as the authors mentioned, there are approaches that scales backpropagation gradients also [Martens and Grosse, 2015; Grosse and Martens, 2016; Ba et al., 2017; George et al., 2018]. Since these methods are highly related to the proposed method, it would be great if the authors could show time complexities and performance differences of these methods as well.

- Experiments on the CIFAR-10 dataset with AlexNet seem not exciting: the proposed Preconditioned approach only outperforms the Fan-out approach marginally. I would say that training a [He et al. 2015]-initialized neural network for 500 more iterations than a preconditioned neural network, yields a similar or better loss.

Overall I think the work is very important and interesting. However, it lacks comprehensive comparison and consideration of more recent neural network layers.

Post Rebuttal Comments
I have read all reviewer comments and the author feedback. I appreciate that the authors addressed the skip connections in Appendix.

1. The authors agree that batch norm requires different initialization schemes that are not included in this paper.
2. I agree with the authors that their approach is complementary to the baseline optimization methods; and both approaches can be applied together. However, I still believe that it is informative to compare the two approaches because: (a). Both approaches address the same problem. Since the optimization based approach adds complexity and computational overhead to implementation, it would be great to show if using the proposed approach eliminates the need for the optimization based approach. (b). Is it necessary to use both approaches, or one of them is good enough?
3. I understand that strong experimental evidence is not always required. However, I believe that the new technical insights of the paper alone is not significant enough (part of the reasons in point 1). Thus I was expecting stronger experimental evidences.

Overall I agree with reviewer 1 that the topic is interesting, but in the paper’s current form, it is not ready. I keep my initial rating of weak reject.

**Experience Assessment:**

I have read many papers in this area.

**Review Assessment: Checking Correctness Of Derivations And Theory:**

I assessed the sensibility of the derivations and theory.

**Review Assessment: Checking Correctness Of Experiments:**

I assessed the sensibility of the experiments.

**Review Assessment: Thoroughness In Paper Reading:**

I read the paper at least twice and used my best judgement in assessing the paper.

---

> ### Author Response · Authors · 2019-11-05
> **Reply**
>
>
> Thanks for the detailed comments.
>
> 1. We address skip connections in detail in the Appendix. In terms of BatchNorm, you are correct that it affects the conditioning, are requires different initialization schemes. The extra analysis needed is significant and we could not fit it into one conference paper. We are looking to address normalization in followup work.
>
> 2. Our initialization method is complementary to the mentioned optimization methods as it can be used together with them.
>
> 3. Only small improvements can be expected by changing initialization methods, it's too much to ask to expect large improvements to final error from a change from arithmetic mean to geometric mean. We are proposing a design principle with theory behind it, the initialization improvements are a secondary result.

---

### Official Review · AnonReviewer1 · 2019-10-27
**Official Blind Review #1**

**Rating:** 3

**Review:**

I think the topic of the paper is interesting, though I think in its current form the paper is not ready.

First of all I find the empirical section quite weak. While the authors attempt to formalize their intuition, as they mention in the work itself, such works are somewhat outside mathematical proof. This is due to the many approximations needed, and assumptions that can not hold in practice. As such the main experimental results in running AlexNet on Cifar-10 and LIBSVM datasets. I think more experimental evidence is needed, e.g. more architectures, more dataset (maybe different data modalities). Is hard to tell from this one main experiment (Cifar 10) to what extend one can trust this initialization.


I think is worth noting that KFAC and (all?) cited methods actually use the Fisher Information matrix (hence being forms of natural gradient) and not the Hessian. The extended Gauss-Newton approximation is indeed the Fisher matrix, I think as discussed in Martens' work (which is heavily cited) though in other works as well. Just as an additional note, the extended Gauss-Newton was introduced in  Schraudolph, N. N. (2002). "Fast curvature matrix-vector products for second-order gradient descent" where it was presented as an approx to the Hessian, this was used by Martens and later the community (and he himself) observed that actually rather than an approx to Hessian this can be thought of as the Fisher Information Matrix.

Relying on the expected squared singular value should be motivated better. The reasoning sounds fine (i.e. minimum and maximum would be too pessimistic) but no data is given. Some statistics over multiple runs. Overall this is a repeating theme in the work. While everything makes sense intuitively, I would have hoped more rigor. More empirical evidence for any such choice. The weight-to-gradient ratio is not a commonly used measure. Maybe show how this ratio progresses over time when training a model. Having multiple runs showing the correlation between things. Table 2 is not referenced in the text. While an average over 10 seeds is provided for Cifar, no error bars.

Overall I think the direction of the work is interesting. And definitely we have not heard the last word on initialization. It plays a crucial role (and indeed bad initialization can easily induce bad local minima (https://arxiv.org/abs/1611.06310)). But I think the paper needs to be written more carefully, with a more thorough empirical exploration, showing different architectures, different datasets. Maybe trying to break the usual initialization, and showing you can do considerably better with newer initialization.

**Experience Assessment:**

I have published one or two papers in this area.

**Review Assessment: Checking Correctness Of Derivations And Theory:**

I assessed the sensibility of the derivations and theory.

**Review Assessment: Checking Correctness Of Experiments:**

I assessed the sensibility of the experiments.

**Review Assessment: Thoroughness In Paper Reading:**

I read the paper at least twice and used my best judgement in assessing the paper.

---

> ### Author Response · Authors · 2019-11-05
> **Reply**
>
> Thank you for the reference on Gauss-Newton methods, we will include this citation.
>
> We would like to disagree respectfully with your assessment that our experimental results are not sufficient. We use 26 different datasets from the LIBSVM repository! These datasets are highly varied in size, dimensionality and difficulty of prediction.  This is far more comprehensive than 90% of published ICLR papers.

---

> > ### Comment · AnonReviewer1 · 2019-11-08
> > **Acknowledgment on size of dataset**
> >
> > Hi,
> >
> > Indeed libsvm seems to contains quite a variation of tasks. I feel like you've done a disservice to the amount of experiments you have done by only having a summary table (table 2) in the main article.
> >
> > That being said, why did you pick this set of tasks, with which I'm not familiar with and have it not seen been used in recent publications. I think picking some more mainstream datasets would again been advantageous for you as it would have easier to evaluate and understand the results.
> >
> > Also since you have these runs, and multiple seeds for cifar, can you grab and show stats that motivate the various choices made in this work? (e.g. not using min/max singular value and so forth).

---

### Decision · Program_Chairs · 2019-12-19

**Decision:**

Reject

**Comment:**

This paper proposes a new design space for initialization of neural networks motivated by balancing the singular values of the Hessian. Reviewers found the problem well motivated and agreed that the proposed method has merit, however more rigorous experiments are required to demonstrate that the ideas in this work are significant progress over current known techniques. As noted by Reviewer 2, there has been substantial prior work on initialization and conditioning that needs to be discussed as they relate to the proposed method. The AC notes two additional, closely related initialization schemes that should be discussed [1,2]. Comparing with stronger baselines on more recent modern architectures would improve this work significantly.

[1]: https://nips.cc/Conferences/2019/Schedule?showEvent=14216
[2]: https://arxiv.org/abs/1901.09321.